# Verifiable Feature Attributions: A Bridge between Post Hoc Explainability and Inherent Interpretability

**Usha Bhalla** [* 1]  **Suraj Srinivas** [* 1]  **Himabindu Lakkaraju** [1]

## Abstract

As machine learning models are increasingly employed in medicine, researchers, healthcare organizations, providers, and patients have all emphasized the need for greater transparency. To provide explanations of models in high-stakes applications, two broad strategies have been outlined in prior literature. Post hoc explanation methods explain the behaviour of complex black-box models by highlighting image regions critical to model predictions; however, prior work has shown that these explanations may not be faithful, and even more concerning is our inability to *verify* them. Specifically, it is nontrivial to evaluate if a given feature attribution is correct with respect to the underlying model. Inherently interpretable models, on the other hand, circumvent this by explicitly encoding explanations into model architecture, making their explanations naturally faithful and verifiable, but they often exhibit poor predictive performance due to their limited expressive power. In this work, we aim to bridge the gap between the aforementioned strategies by proposing *Verifiability Tuning* (VerT), a method that transforms black-box models into models with verifiable feature attributions. We begin by introducing a formal theoretical framework to understand verifiability and show that attributions produced by standard models cannot be verified. We then leverage this framework to propose a method for building verifiable models and feature attributions from black-box models. Finally, we perform extensive experiments on semi-synthetic and real-world datasets, and show that VerT produces models (1) yield explanations that are correct and verifiable and (2) are faithful to the original black-box models they are meant to explain.

---

[*]Equal contribution  [1]Harvard University, Cambridge, MA. Correspondence to: Usha Bhalla <usha_bhalla@g.harvard.edu>, Suraj Srinivas <ssrinivas@seas.harvard.edu>.

*Proceedings of the $40^{th}$ International Conference on Machine Learning*, Honolulu, Hawaii, USA. PMLR 202, 2023. Copyright 2023 by the author(s).

## 1. Introduction

The rapid adoption of machine learning in many high-stakes applications, such as healthcare, finance, and more, has brought with it the rise in popularity of explainable AI (XAI), which attempts to demystify ML models so that users can make informed decisions about their trustworthiness, accuracy, and usefulness. XAI methods can be broadly split into two categories: post hoc explainability and inherent interpretability. The majority of post hoc explanations aim to explain instance-level decisions of pre-trained black box models through feature attribution, or labelling which features were the most relevant to a model's decision. These methods often work by approximating model behavior through local linear function approximation (Han et al., 2022). However, given that these methods are approximations at the local input regions, they are often not *faithful* to the true behavior of models beyond the local regions. This is particularly concerning because it is difficult to directly and decisively verify the faithfulness or correctness of the attributions produced. Inherently interpretable models, on the other hand, are explainable by nature and are crafted such that humans can clearly trace the steps, reasoning, and computations made by a model. As such, the explanations yielded by these models are highly faithful and can be readily verified; however, the models themselves are often less useful in practice due to their decreased performance and limited expressivity.

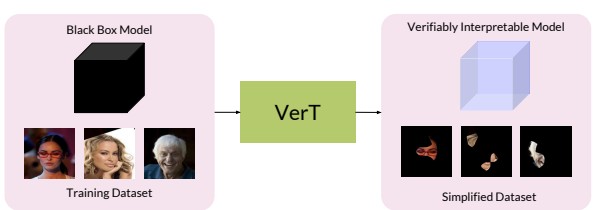

Figure 1: Illustration of our method, Verifiability Tuning, which converts black-box models and their training datasets into verifiably-interpretable models and the minimal dataset required to yield them.

In this work, we aim to bridge the gap between the two aforementioned categories of methods by proposing *Veri-*

*fiability Tuning* (VerT), which converts black box models into verifiably-interpretable ones. Specifically, the models produced by our method generate feature attributions that are verifiable by nature and highly faithful to the underlying behavior of the original black-box model. Our contributions are the following:

1. We first formalize a framework for understanding feature attribution verification, and show theoretically that feature attributions applied to black-box models are unverifiable.

2. We propose *verifiability tuning* (VerT), a novel method that converts black-box models into verifiably-interpretable ones, such that the feature attributions of the resulting models are verifiable, in that model predictions remain consistent after masking unimportant features.

3. We perform experiments on semi-synthetic and real-world computer vision and medical imaging datasets and show that VerT outputs models that are faithful to its original input models, and that their feature attributions are interpretable, verifiable, and correct, in the sense that they match ground-truth feature attributions.

By proposing a method to produce *verifiable* and *faithful* attributions while still leveraging black-box models, our work offers a promising approach to address the limitations of both post hoc explanation methods and inherently interpretable models.

## 2. Related Work

**Post-Hoc Explainability.** Post-hoc explainability methods aim to explain the outputs of fully trained black-box models either on an instance-level or global level. The most common post-hoc methods are feature attribution methods that rank the relative importance of features, either by explicitly producing perturbations (Ribeiro et al., 2016; Lundberg & Lee, 2017), or by computing variations of input gradients (Smilkov et al., 2017; Srinivas & Fleuret, 2019; Selvaraju et al., 2017). Perturbation-based methods are especially popular in computer vision literature (Zeiler & Fergus, 2014; Fong & Vedaldi, 2017; Fong et al., 2019; Dabkowski & Gal, 2017), which use feature removal strategies adapted specifically for image data. However, these methods all assume a specific form for feature removal, and we show theoretically in Section 3 that this can lead to unverifiable attributions.

**Inherently Interpretable Models.** Inherently interpretable models are constructed such that we know exactly what they do, either through their weights or explicit modular reasoning. As such, the explanations provided by these

models are more accurate than those given by post-hoc methods; however, the performance of interpretable models often suffers when compared to unconstrained black-box architectures. The most common inherently interpretable model classes include linear models, decision trees and rules with limited depth, GLMs, GAMs (Hastie, 2017), JAMs (Chen et al., 2018; Yoon et al., 2019; Jethani et al., 2021), and prototype- and concept-based models (Chen et al., 2019; Koh et al., 2020). While (Chen et al., 2019; Koh et al., 2020) leverage the expressivity of deep networks, they constrain hypothesis classes significantly and still often suffer from a decrease in performance. Among these, our work most closely relates to JAMs, which amortise feature attribution generation using a learnt masking function to generate attributions in a single forward pass, and trains black-box models using input dropout. On other hand, JAMs (1) trains models from scratch, whereas VerT can interpret black-box models, (2) amortises feature attributions using a masking function resulting in less accurate attributions, (3) trains models to be robust to a large set of candidate masks via input dropout, leading to low predictive accuracy, whereas VerT trains models only to be robust to the optimal mask, leading to more flexibility and higher predictive accuracy.

**Evaluating Correctness of Explanations.** As explainability methods grow in number, so does the need for rigorous evaluation of each method. Research has shown that humans naively trust explanations regardless of their "correctness" (Lakkaraju & Bastani, 2020), especially when explanations confirm biases or look visually appealing. Common approaches to evaluate explanation correctness rely on feature / pixel perturbation (Samek et al., 2016; Srinivas & Fleuret, 2019; Agarwal et al., 2022), i.e., an explanation is correct if perturbing unimportant feature results in no change of model outputs, whereas perturbing important features results in large model output change. Hooker et al. (2019) proposed remove-and-retrain (ROAR) for evaluating feature attribution methods by training surrogate models on subsets of features denoted un/important by an attribution method and found that most gradient-based methods are no better than random. While prior works focused on developing metrics to evaluate correctness of explanations, our method VerT produces models that have explanations that are accurate by design, according to pixel-perturbation methods.

## 3. A Framework for Verifiability

In this section, we first demonstrate that the attributions produced by standard models are difficult to verify. We then introduce formal notions of verifiable models and feature attributions and characterize the conditions under which such verifiability can be achieved.

Intuitively, feature attribution methods work by simulating the removal of certain features and estimating how the model behaves when those features are removed: removal of unimportant features should not change model behaviour. Typically, this removal is implemented in practice by replacing features with scalar values, such as the mean of the dataset (Zeiler & Fergus, 2014; Samek et al., 2016; Srinivas & Fleuret, 2019). However, this can result in out-of-distribution inputs that the classifier mishandles, making it challenging to verify whether the classifier is solely reliant on the signal from the relevant features. To ground this argument in an example, consider a model that classifies cows and camels. For an image of a camel, a feature attribution might note that only the hump of camel and the sand it stands on are important for classification. As such, we would expect that the sky was irrelevant to the classifier's prediction, and we can concretely test this by altering it and creating a counterfactual sample. For example, we could mask the sky with an arbitrary uniform color; however, this may result in the sample being out-of-distribution for the model, and its prediction may change drastically even if the sky was actually not important for prediction. Ideally, we would be able to mask the sky in a manner that preserves the on-manifoldness of the image, but this is extremely tricky and dependent on both the dataset and the model. We formalize this argument below by defining feature attributions with respect to a given counterfactual distribution of masks that determine feature replacement, which we refer to as $(\epsilon, \mathcal{Q})$-feature attribution.

**Notation.** Throughout this paper, we shall assume the task of classification with inputs $\mathbf{x} \sim \mathcal{X}$ with $\mathbf{x} \in \mathbb{R}^d$ and $y \in [1, 2, ...C]$ with $C$-classes. We consider the class of deep neural networks $f : \mathbb{R}^d \to \triangle^C$ which map inputs $\mathbf{x}$ onto a $C$-class probability simplex. This paper considers binary feature attributions, which are represented as binary masks $\mathbf{m} \in \{0, 1\}^d$, where $\mathbf{m}_i = 1$ indicates an important feature and $\mathbf{m}_i = 0$ indicates an unimportant feature. We shall also use the notation $\mathbf{m}' \subset \mathbf{m}$ to represent $\mathbf{m}'$ such that $\mathbf{m}'_i = 1$ on a subset of indices $i$ for which $\mathbf{m}_i = 1$. We say that $\mathbf{m}'$ is a *subset mask* of $\mathbf{m}$.

### 3.1. Verifiable Models and Feature Attributions

We first define the notion of feature attribution in the following manner, where the feature replacement method is made explicit, and features are replaced with samples from a counterfactual distribution $\mathcal{Q}$. Particularly, we are interested in binary attributions as opposed to real-valued attributions because of their greater interpretability (i.e., a feature is either considered important for prediction or not rather than somewhat important).

*Definition* 1. $(\epsilon, \mathcal{Q})$-feature attribution (QFA) is a binary mask $\mathbf{m}(f, \mathbf{x}, \mathcal{Q})$ that relies on a model $f(\cdot)$, an instance $\mathbf{x}$,

and a $d$-dimensional counterfactual distribution $\mathcal{Q}$, and is given by

$$\mathbf{m}(f, \mathbf{x}, \mathcal{Q}) = \arg \min_{\mathbf{m}'} \|\mathbf{m}'\|_0$$

$$\text{such that} \quad \mathbb{E}_{q \sim \mathcal{Q}} \|f(\mathbf{x}_s(\mathbf{m}', q)) - f(\mathbf{x})\|_1 \leq \epsilon$$

$$\text{where} \quad \mathbf{x}_s(\mathbf{m}, q) = \mathbf{m} \odot \mathbf{x} + (1 - \mathbf{m}) \odot q$$

Thus an $(\epsilon, \mathcal{Q})$-feature attribution (henceforth, *QFA*) refers to the sparsest mask that can be applied to an image such that the model's output remains approximately unchanged. Observe that QFA depends on the feature replacement distribution $\mathcal{Q}$, where $\mathcal{Q}$ is independent of both $\mathbf{x}$ and $y$. This generalizes the commonly used heuristics of replacing unimportant features with the dataset mean, in which case $\mathcal{Q}$ is a dirac delta distribution at the mean value. The choice of $\mathcal{Q}$ is indeed critical, as an incorrect choice can hurt our ability to recover the correct attributions due to the resulting inputs being out-of-distribution, and the classifier being sensitive to such changes. Specifically, an incorrect $\mathcal{Q}$ can result in QFA being less sparse, as masking even a few features with the wrong $\mathcal{Q}$ would likely cause large deviations in the model's outputs. As a result, given a model, we must aim to the find the $\mathcal{Q}$ that leads to the sparsest QFA masks. However, the problem of searching over $\mathcal{Q}$ is complex, as it requires searching over the space of all $d$-dimensional distributions. To avoid this, we reverse the problem: given $\mathcal{Q}$, we find the class of models which have the sparsest QFAs w.r.t. that particular $\mathcal{Q}$. We call this the $\mathcal{Q}$-verifiable model class, which we define below:

*Definition* 2. $\mathcal{Q}$-verifiable model class $\mathcal{F}_v(\mathcal{Q})$: For some given distribution $\mathcal{Q}$, the class of models $\mathcal{F}$ for which $\mathcal{Q}$ gives the sparsest QFA mask as opposed to any other $\mathcal{Q}'$, such that for all $f \in \mathcal{F}$,

$$\mathcal{Q} = \arg \min_{\mathcal{Q}'} \mathbb{E}_{\mathbf{x}} \|\mathbf{m}(f, \mathbf{x}, \mathcal{Q}')\|_0$$

is called the model class with verifiable QFA.

For the rest of this paper, we shall refer to QFA applied to a model from a $\mathcal{Q}$-verifiable model class as a "verifiable" feature attribution.

### 3.2. Recovering the Signal-Distractor Decomposition

In the study of feature attribution, the 'ground truth' attributions are often unspecified. Here, we show that for datasets that are signal-distractor decomposable, formally defined below, there exists a ground truth attribution, and feature attributions for optimal verifiable models are able to recover it. Intuitively, given an object classification dataset between cows and camels, the "signal" refers to the regions in the image that are discriminative, or correlated with the label, such as the cows or camels. The distractor refers to everything

else, such as the background or sky. Note that if objects in the background are correlated with the label, i.e. sand or grass, those would be part of the signal, not the distractor. For all datasets with this decomposition, the ground truth attributions correspond to the signal. We first begin by formally defining the signal-distractor decomposition.

*Definition* 3. A labelled dataset $D = \{(\mathbf{x}, y)_{i=1}^{N}\}$ is said to be signal-distractor decomposable if its generative process is given by:

$$\mathbf{s} \sim \mathcal{X}_{\text{signal}}(y) \; ; \; \mathbf{d} \sim \mathcal{X}_{\text{distractor}} \; ; \; \mathbf{m} \sim \mathcal{M}$$
$$\mathbf{x} = \mathbf{s} \odot \mathbf{m} + \mathbf{d} \odot (1 - \mathbf{m})$$

where the mask $\mathbf{m} \in \{0, 1\}^d$ and the distractor $\mathbf{d} \in \mathbb{R}^d$ are independent of the label $y$, and the signal $\mathbf{s} \in \mathbb{R}^d$ depends on $y$, with the following conditions met: (1) $p(y \mid \mathbf{x}) = p(y \mid \mathbf{s} \odot \mathbf{m}) = p(y \mid \mathbf{s})$, and (2) for any other mask $\mathbf{m} \not\subset \mathbf{m}'$, we have $p(y \mid \mathbf{s} \odot \mathbf{m}') < p(y \mid \mathbf{s} \odot \mathbf{m})$

We have the condition here that any mask that is not a superset of the ground truth mask leads to a strict loss in predictive power of the input. It follows from this non-redundancy constraint that given the distractor $\mathbf{d}$ and the signal $\mathbf{s}$, the mask that determines $\mathbf{x}$ is unique, and is the sparsest mask such that condition (1) is met.

**Theorem 3.1.** *For datasets which are signal-distractor decomposable, QFA recovers the signal distribution when applied to the optimal predictor $f_v^* \in \mathcal{F}_v(\mathcal{Q})$.*

*Proof Idea.* For datasets with the signal-distractor decomposition, the optimal $\mathcal{Q}$ is always equal to the ground truth distractor, and this leads to the sparsest QFA. If a $\mathcal{Q}$-verifiable model aims to recover the sparsest masks, then its QFA mask must equal that obtained by setting $\mathcal{Q}$ equal to the distractor. From uniqueness arguments, this is possible only when the signal distribution is recovered by QFA, such that it is not masked by $\mathcal{Q}$. □

*Corollary.* For datasets which are signal-distractor decomposable, QFA does not recover the signal distribution when applied to a predictor $f \notin \mathcal{F}_v(\mathcal{Q})$.

This follows from the fact that for any $f \notin \mathcal{F}_v(\mathcal{Q})$, there exists some other $\mathcal{Q}'$ that results in a sparser mask, indicating that the true signal distribution is not recovered. Thus, this shows that feature attributions applied to the incorrect model class can be less effective - in this case they fail to recover the ground truth signal distributions.

To summarize, we have defined a feature attribution method with the feature removal process made explicit via the counterfactual distribution $\mathcal{Q}$. To minimize sparsity of the attribution masks, we have to either (1) find the best distribution $\mathcal{Q}$, which is difficult to compute, or (2) given a fixed $\mathcal{Q}$, use models from the $\mathcal{Q}$-verifiable model class $\mathcal{F}_v(\mathcal{Q})$. Finally,

we find that feature attributions derived from model class $\mathcal{F}_v(\mathcal{Q})$ are able to recover the signal-distractor decomposition of datasets.

## 4. Verifiability Tuning

In the previous section we showed that given a $\mathcal{Q}$-verifiable model $f_v \in \mathcal{F}_v(\mathcal{Q})$, we are able to apply QFA to recover the ground truth signal from the dataset. In this section, we shall discuss how to practically build such verifiable models that recover the optimal ground truth signal, $\mathbf{x}_s(\mathbf{m}, q) = \mathbf{x} \odot \mathbf{m}(\mathbf{x}) + q \odot (1 - \mathbf{m}(\mathbf{x}))$, given a pre-defined counterfactual distribution $q \sim \mathcal{Q}$ that determines feature replacement.

**Relaxing QFA.** We note that QFA as defined in definition 1 is difficult to optimize in its current form due to its use of $\ell_0$ regularization and its constrained form. To alleviate this problem, we perform two relaxations: first, we relax the $\ell_0$ objective into an $\ell_1$ objective, and second, we convert the constrained objective to an unconstrained one by using a Lagrangian. The resulting objective function is given in equation 1. Assuming the model $f_v$ is known to us, we can minimize this objective function to obtain $(\epsilon, \mathcal{Q})$-feature attributions for each point $\mathbf{x} \in \mathcal{X}$.

$$\mathcal{L}_{\text{QFA}}(\theta, \{\mathbf{m}(\mathbf{x})\}_{\mathbf{x} \in \mathcal{X}}) =$$
$$\mathbb{E}_{\mathbf{x} \in \mathcal{X}} \left[ \underbrace{\|\mathbf{m}(\mathbf{x})\|_1}_{\text{mask sparsity}} + \lambda_1 \underbrace{\|f_v(\mathbf{x}; \theta) - f_v(\mathbf{x}_s(\mathbf{m}, q)); \theta)\|_1}_{\text{data distillation}} \right] \tag{1}$$

**Enforcing Model Verifiability via Distillation.** Assuming that the optimal masks denoting the signal-distractor decomposition are known w.r.t. every training data point (i.e., $\{\mathbf{m}(\mathbf{x})\}_{\mathbf{x} \in \mathcal{X}}$), one can project any black-box model into a $\mathcal{Q}$-verifiable model via distillation. Specifically, we can use equation 2 for this purpose, which contains (1) a data distillation term to enforce the $\epsilon$ constraint in QFA, and (2) a model distillation term to enforce that the resulting model and original model are approximately equal. Accordingly, the black-box model $f_b$ and our resulting model $f_v$ both have the same model architecture, and we initialize $f_v = f_b$.

$$\mathcal{L}_{\text{train}}(\theta, \{\mathbf{m}(\mathbf{x})\}_{\mathbf{x} \in \mathcal{X}})$$

$$= \underset{\mathbf{x} \in \mathcal{X}}{\mathbb{E}} \left[ \underbrace{\|f_v(\mathbf{x}; \theta) - f_v(\mathbf{x}_s(\mathbf{m}(\mathbf{x}), q)); \theta)\|_1}_{\text{data distillation}} \right]$$

$$+ \underset{\mathbf{x} \in \mathcal{X}}{\mathbb{E}} \left[ \lambda_2 \underbrace{\|f_b(\mathbf{x}) - f_v(\mathbf{x}; \theta)\|_1}_{\text{model distillation}} \right] \quad (2)$$

**Alternating Minimization between $\theta$ and $\mathbf{m}$.** We are interested in both of the above objectives: we would like to recover the optimal masks from the dataset, as well as use those masks to enforce $(\epsilon, \mathcal{Q})$ constraints via distillation to yield our $\mathcal{Q}$-verifiable models. We can thus formulate the overall optimization problem as the sum of these terms, as shown in equation **??**. Notice that both these objectives assume that either the optimal masks, or the verifiable model is known, and in practice we know neither. A common strategy in cases which involve optimizing over multiple sets of variables is to employ alternating minimization (Jain & Kar, 2017), which involves repeatedly fixing one of the variables and optimizing the other. We handle the constrained objective on the mask variables via projection, i.e., using hard-thresholding / rounding to yield binary masks.

$$\theta^*, \{\mathbf{m}^*(\mathbf{x})\}$$
$$= \arg \min_{\theta, \mathbf{m}} (\mathcal{L}_{\text{train}}(\theta, \{\mathbf{m}(\mathbf{x})\}) + \mathcal{L}_{\text{QFA}}(\theta, \{\mathbf{m}(\mathbf{x})\}))$$

$$\text{such that} \quad \mathbf{m}(\mathbf{x}) \in \{0, 1\}^d \quad \forall \mathbf{x} \in \mathcal{X}$$

**Iterative Mask Rounding with Increasing Sparsity.** In practice, mask rounding makes gradient-based optimization unstable due to sudden jumps in the variables induced by rounding. This problem commonly arises when dealing with sparsity constraints. To alleviate this problem, use a heuristic that is common in the model pruning literature (Han et al., 2015) called iterative pruning, which involves introducing a rounding schedule, where the sparsity of the mask is gradually increased during optimization steps. Inspired from this choice, we employ a similar strategy over our masks variables.

**Practical Details.** We implement these objectives as follows. First, we initialize the verifiable model to be the same as the original model, $f_v = f_b$, and the mask to be all ones, $D_s = m \odot D_d, m = 1$. We then iteratively (1) simplify $D_s$ by optimizing $\mathcal{L}_{QFA}$ until $\mathbf{m}$ converges, (2) round $\mathbf{m}$ such that it is binary (i.e. $D_s$ is a subset a features in $D_d$ rather than a weighting of them), and (3) update $f_v$ by minimizing $\mathcal{L}_{train}$ such that $D_s$ is equally as informative as $D_d$ to $f_v$ and $f_v$ is functionally equivalent to $f_b$. As per

---

**Algorithm 1** Verifiability Tuning

**Input:** Dataset $D_d := (x, y)$, model $f_b$, hyperparameter $k$ rounding steps
**Hyperparameters:** $k$ rounding steps, $u$ mask scaling factor
$\{\mathbf{m}(\mathbf{x})\}$, s.t. $\mathbf{m}(\mathbf{x}) \leftarrow$ ones with shape $\mathbb{R}^{d/u}$
$f_v \leftarrow f_b$
**for** $k$ rounding steps **do**
  **while** $\mathcal{L}_{\text{QFA}}$ not converged **do**
    $\{\mathbf{m}(\mathbf{x})\} \leftarrow \{\mathbf{m}(\mathbf{x})\} + \nabla_{\mathbf{m}} \mathcal{L}_{\text{QFA}}$
  **end while**
  $\mathbf{m} \leftarrow \text{round}(\mathbf{m}) \quad \forall \mathbf{m} \in \{\mathbf{m}(\mathbf{x})\}$
  **while** $\mathcal{L}_{\text{train}}$ not converged **do**
    $f_v \leftarrow f_v + \nabla_\theta \mathcal{L}_{\text{train}}$
  **end while**
**end for**
**return** $\{\mathbf{m}(\mathbf{x})\}, f_v$

---

Definition 2, we replace masked pixels in $D_s$ with a predetermined counterfactual distribution $\mathcal{Q}$. This ensures that the given $\mathcal{Q}$ is the optimal counterfactual distribution for $f_v$, meaning $f_v$ comes from the $\mathcal{Q}$-verifiable model class $\mathcal{F}_v(\mathcal{Q})$. Pseudocode is shown below.

**Mask Scale.** In order to encourage greater "human interpretability," we explore different levels of mask granularity. We do this by downscaling the masks before optimization. Concretely, we initialize the masks to be of size $m_d = x_d/u$, where $x_d$ is the dimension of the image $\mathbf{x}$ and $u$ is the superpixel size we wish to consider. We then upsample the mask to be of dimension $x_d$ before applying it to $\mathbf{x}$. The more we downscale the mask by (i.e. the greater $u$ is), the larger the superpixels, or features, of $\mathbf{x}_s$ are, and the more interpretable and visually cohesive the distilled dataset $\mathbf{x}_s$ is.

## 5. Experimental Evaluation

In this section, we present our empirical evaluation in detail. We consider various quantitative and qualitaive metrics to evaluate the correctness of feature attributions given by VerT models as well as the faithfulness of VerT models to the models they are meant to explain. We also evaluate VerT models' ability to explain models manipulated to have arbitrary uninformative input gradients. Finally, we analyze the effect of the mask downscaling hyperparameter on attributions.

**Datasets.** **Hard MNIST:** The first is a harder variant of MNIST where the digit is randomly placed on a small subpatch of a colored background. Each sample also contains small distractor digits and random noise patches. For this dataset, we consider the signal to be all pixels contained

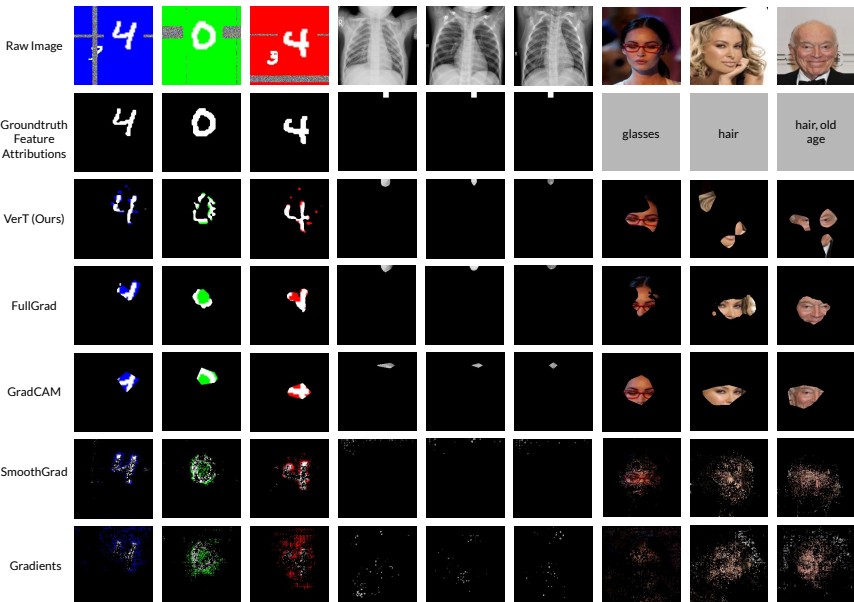

Figure 2: Visualization of datasets and attribution methods considered in this work. *Row 1*: Raw data samples, *Row 2*: Ground truth attributions, *Row 3*: VerT attributions, *Rows 4-7*: Baseline methods.

within the large actual digit, and the distractor to be all pixels in the background, noise, and smaller digits. **Chest X-ray:** Second, we consider a semi-synthetic chest x-ray dataset for pneumonia classification (Kermany et al., 2018). To control exactly what information the model leverages such that we can create ground truth signal-distractor decompositions, we inject a spurious correlation into this dataset. We place a small, barely perceptible noise patch on random location at the top of each image in the "normal" class. We confirm that the model relies only on the spurious signal during classification by testing the model's drop in performance when flipping the correlation (adding the noise patches to the "pneumonia" class) and seeing that the accuracy goes from 100% to 0%. As such, for this dataset, the signal is simply the noise patch and the distractor is the rest of the xray. **CelebA:** The last dataset is a subset of CelebA (Liu et al., 2018) for hair color classification with classes {dark hair, blonde hair, gray hair}. We correlate the dark hair class with glasses to allow for qualitative evaluation of each methods' ability to recover spurious correlations. This dataset does not have a ground truth signal distractor decomposition, as there are many unknown discriminative spurious correlations the model may rely upon.

**Models.** For all experiments, we use ImageNet pre-trained ResNet18s for our baseline models, $f_b$. All models achieve over 96% test accuracy. We train VerT models with $Q \sim \mathbb{1}_{d*d} * \mathcal{N}(\mu(D_d), \sigma^2(D_d))$, meaning that each image is masked with a uniform color drawn from a normal distribution around the mean color of the dataset. For all

evaluation, we use $Q \sim \mathbb{1}_{d*d} * \mathcal{N}(\mu(D_d), 0)$ (the dirac delta of the dataset mean) to ensure that masked samples are minimally out-of-distribution for the baseline models $f_b$.

### 5.1. Evaluating the Correctness of Feature Attributions

**Pixel Perturbation Tests.** We test the verifiability of our explanations via the pixel perturbation variant proposed in (Srinivas & Fleuret, 2019; Hooker et al., 2019), where we mask the $k$ least salient pixels as determined by any given attribution method and check for degradation in model performance with the mean of the dataset. This metric evaluates whether the $k$ masked pixels were necessary for the model to make its prediction. As mentioned in previous works, masked samples come from a different distribution than the original samples, meaning poor performance after pixel perturbation can either be a product of the model's performance on the masks or the feature attribution scores being incorrect. To disentangle the correctness of the attributions from the verifiability of the model, we perform pixel perturbation tests on ground truth feature attributions, with results reported in the appendix. Note that our method returns binary masks, but this metric requires continuous valued attributions to create rankings. As such, for this experiment we use the attributions created by our method *before rounding*.

Results are shown in Figure 3. We find that the attributions produced by VerT , used in conjunction with VerT models outperform all baselines. Furthermore, VerT attributions tested on the baseline model also generally perform better

than gradient-based attributions.

Table 1: Intersection Over Union Results

|  | Hard MNIST | Chest X-ray |
|---|---|---|
| VerT (Ours) | **0.461** ±0.08 | **0.821**± 0.05 |
| SmoothGrad | 0.252±0.05 | 0.045 ±0.05 |
| GradCAM | 0.295 ±0.09 | 0.000 ±0.00 |
| Input Grad | 0.117 ±0.05 | 0.017±0.03 |
| FullGrad | 0.389 ±0.10 | 0.528 ±0.12 |

**Intersection Over Union.**    We further evaluate the correctness of our attributions by measuring their Intersection Over Union (IOU) with the "ground truth" attributions. We use the signal from the ground truth signal-distractor decomposition as described in 5 for the ground truth attributions. For each image, if the ground truth attribution is comprised of $n$ pixels, we take the intersection over union of the top $n$ pixels returned by the explanation method and the $n$ ground truth pixels, meaning an IOU of 1 corresponds to a perfectly aligned/correct attribution. Results are shown in 1, where our method performs the best for both datasets. We report mean and standard deviation over the dataset for each method.

### 5.2. Evaluating the Faithfulness of VerT Models

To ensure that the VerT model ($f_v$) returned by our method is faithful the the original model ($f_b$) it approximates, we test the accuracy of VerT models with respect to the predictions produced by the original model. Specifically, we take the predictions of the original model $f_b$ to be the labels of the dataset. We evaluate VerT models on both the original dataset ($f_v(D_d) \approx f_b(D_d)$) and the simplified dataset ($f_v(D_s) \approx f_b(D_d)$), with results shown in 2. We see that VerT models are highly faithful to the baseline models they approximate.

### 5.3. Qualitative Analysis of Simplified Datasets

We first explore how well VerT recovers signal-distractor decompositions. For CelebA, we see that all methods except for input gradients correctly recover the spurious correlation for the "dark hair/glasses" class, however only our method provides useful insights into the other two classes. We see that our method correctly identifies hair as the signal for the "blonde hair" class, whereas other methods simply look at the eyes, which are not discriminative. Furthermore, we see that for the "gray hair" class, our method picks up on hair, as well as initially unknown spurious correlations such as wrinkles and bowties. For Hard MNIST, we see that our method clearly isolates the signal and ignores the distractor.

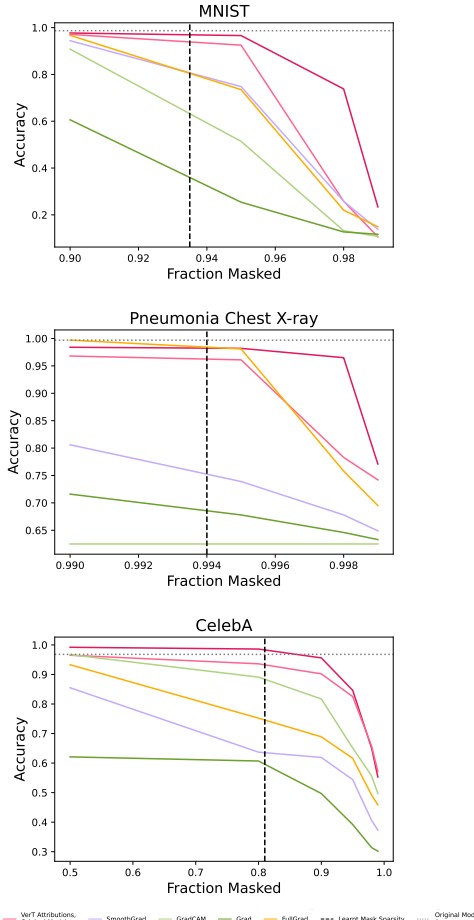

Figure 3: Pixel perturbation tests (higher is better) for MNIST, Chest X-ray, and CelebA. VerT's recommended mask sparsity is shown as a vertical dashed line. We observe that VerT performs the best overall. Refer to Section 5.1 for details.

FullGrad and GradCAM suffer from a locality bias and tend to highlight the center of each digit. SmoothGrad and vanilla gradients are much noisier and highlight edges and many random background pixels. For the Chest X-ray dataset, we see that our method and FullGrad perfectly highlight the spurious signal. GradCAM again suffers from a centrality bias, and cannot highlight pixels on the edge. SmoothGrad and gradients appear mostly random to the human eye.

We also consider the visual quality of our attributions compared with the baselines (examples are shown in 2). We find that our method, FullGrad, and GradCAM appear the most interpretable, as opposed to SmoothGrad and vanilla gradients, because they consider features at the superpixel level rather than individual pixels. We also see that GradCam and FullGrad seem relatively class invariant, and tend to focus on the center of most images, rather than the discriminative features for each class, providing for less useful insights into the models and datasets.

Table 2: Faithfulness of VerT to Original Model

|  | Hard MNIST | Chest X-ray | CelebA |
|---|---|---|---|
| Faithfulness on Original Data | 0.996 | 1.00 | 0.995 |
| Faithfulness on Simplified Data | 0.987 | 1.00 | 0.975 |

### 5.4. Robustness to Adversarial Manipulation of Explanations

In this section, we highlight our method's robustness to adversarial explanation manipulation. To this end, we follow the manipulation proposed in (Heo et al., 2019), which adversarially manipulates gradient-based explanations. This is achieved by adding an extra term to the training objective that encourages input gradients to equal an arbitrary uninformative mask of pixels in the top left corner of each image. Note that model accuracy on the classification task is the same as training with only cross-entropy loss.

We repeat experiments for all evaluation metrics on these manipulated models, with pixel-perturbation shown below 7, and IOU, model faithfulness, and model verifiability in the appendix. We see that gradient-based methods perform significantly worse on manipulated models; however, our method remains relatively invariant. We also note while the models are only manipulated to have arbitrary input gradients, SmoothGrad and GradCAM are also heavily affected such that their attributions are entirely uninterpretable as well, as shown below.

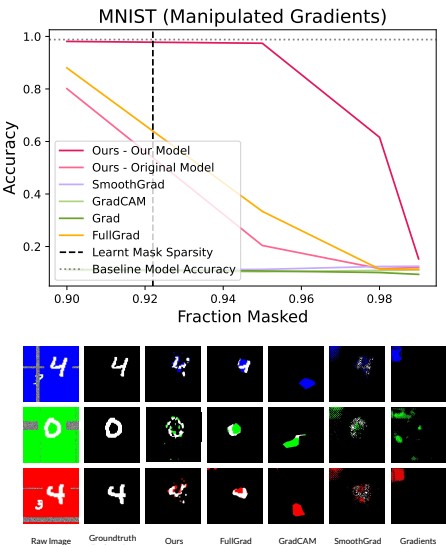

Figure 4: Pixel perturbation test and example images for models trained with gradient manipulation on Hard MNIST. Results for other datasets are in the appendix. Refer to Section 5.4 for details.

### 5.5. Attribution Sensitivity to Hyperparamaters

We conduct an ablation study on the choice of how much to downscale the mask by. The less we downscale by, the more fine-grained the mask is, allowing for optimization over a larger set of masks. However, the more we downscale by, the visually cohesive or "interpretable" to humans the masks are. We evaluate the trade-off between these two via pixel perturbation tests over multiple downscaling factors and with qualitative evaluations of the final masks in 5. We see that a downscaling factor of 8 performs the best on pixel perturbation tests. Increased factors of downscaling impose a greater locality constraint that results in informative pixels being masked, as shown in the visualization.

## 6. Discussion

In this paper, we seek to bridge the gap between existing post hoc explanation methods, which are unfaithful and non-verifiable, and inherently interpretable models, which are restrictive and often less performant than black-box models. In particular, we propose, VerT, a method that can transform any black-box model into a verifiably interpretable model, whose feature attributions can be easily evaluated for faithfulness. We empirically evaluate VerT and find that the resulting verifiable models are highly faithful and produce interpretable and verifiable attributions.

**Limitations.** We note that VerT requires full access to the training dataset and the baseline model. Furthermore,

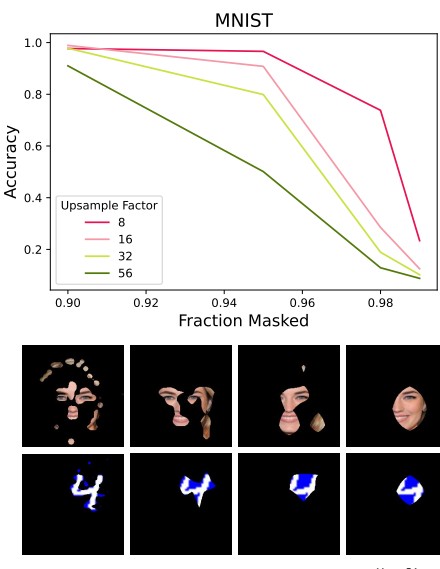

Figure 5: Ablation study of pixel perturbation test with varying levels of mask downscaling for MNIST. Example images for both datasets are shown below.

while it produces verifiable feature attributions that tell us how important each feature is to the model's prediction, it does not tell us what the relationship between important features and the output/label is, as is true with all feature attributions. Finally, the utility of these attributions, and indeed, all feature attributions, relies on the existence of a non-trivial signal distractor decomposition of the dataset. If such a decomposition does not meaningfully exist, for example when the entire input is the signal, then ours, and all other feature attributions, are inapplicable.

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

## A. Proofs

**Theorem A.1.** *For datasets with signal-distractor decompositions, $\epsilon\mathcal{Q}$-feature attributions applied to $\epsilon\mathcal{Q}$ verifiable models, $f_v \in \mathcal{F}_v(\mathcal{Q})$, recover the signal distribution for the optimal predictor $f_v^*$.*

*Proof.* We first notice that the ideal counterfactual distribution $\mathcal{Q}_{opt}$ is exactly the distractor distribution, i.e., if we set $\mathcal{Q}_{opt} = \mathcal{X}_{\text{distractor}}$, we recover the inputs $\mathbf{x}$, if we know the ground truth mask $\mathbf{m}_{\text{dataset}}$, i.e, $\mathbf{x} \odot \mathbf{m}_{\text{dataset}} = \mathbf{s} \odot \mathbf{m}_{\text{dataset}}$, which implies that the "simplified" inputs equal the real inputs, i.e., $\mathbf{x}_s = \mathbf{x} \odot \mathbf{m}_{\text{dataset}} + \mathbf{q}_{opt} \odot (1 - \mathbf{m}_{\text{dataset}}) = \mathbf{s} \odot \mathbf{m}_{\text{dataset}} + \mathbf{d} \odot (1 - \mathbf{m}_{\text{dataset}}) = \mathbf{x}$. However we do not know the ground truth mask $\mathbf{m}_{\text{dataset}}$.

Given the "non-redundancy" constraint in the signal-distractor decomposition, it follows that the ground truth mask is the sparsest mask such that $p(y \mid \mathbf{s} \odot \mathbf{m}) = p(y \mid \mathbf{x})$, or $f^*(\mathbf{x}) = f^*(\mathbf{x}_s)$, since $f^*$ is the optimal predictor. Given the uniqueness of the ground truth mask by definition, we recover $\mathbf{m}_{\text{dataset}} = \mathbf{m}_{\text{QFA}}$ where $\mathbf{m}_{\text{QFA}}$ is computed using $\mathcal{Q}_{opt}$.

For a "non-ideal" fixed distractor $\mathcal{Q}$, and the corresponding verifiable model class $\mathcal{F}_v(\mathcal{Q})$, we know that $\mathcal{Q}$ attains least sparsity compared to any other counterfactual. However, we know from the above argument that $\mathcal{Q}_{opt} = \mathcal{X}_{\text{distractor}}$ attains the least sparsity for any optimal predictor. Thus it follows that these must be equal. This implies that the mask recovered by $\mathbf{m}_{\text{QFA}} = \mathbf{m}_{\text{dataset}}$. $\square$

We now present proof for an additional statement not described in the main text, where we connect QFA to other commonly used feature attributions via the local function approximation framework (Han et al., 2022) as follows.

**Theorem A.2.** *QFA is an instance of the local function approximation framework (LFA), with (1) random binary perturbations, and (2) an interpretable model class consisting of linear models with binary weights*

*Proof.* Assume a black box model given by $f_b(\mathbf{x}; \mathbf{m}) = \mathbb{1}\left(\mathbb{E}_q \|f(\mathbf{x}_s(\mathbf{m}, q)) - f(\mathbf{x})\|_2 \leq \epsilon\right)$, loss function $\ell(f, g, x, \xi) = (f(x; \xi) - g(\xi))^2$, neighborhood perturbation $Z = \text{Uniform}(0, 1)^d$, and an interpretable model family $\mathcal{G}$ being the class of binary linear models.

For these choices, it is easy to see that

$$\arg\min_{g \in \mathcal{G}} \ell(f, g, \mathbf{x}, \xi)$$

$$= \arg\min_{g \in \mathcal{G}} \mathbb{E}_\xi \left(f_b(\mathbf{x}; \xi) - g^\top \xi\right)^2 + \lambda \|g\|_0$$

$$= \arg\min_{g \in \mathcal{G}} \mathbb{E}_\xi \left(\mathbb{1}\left(\mathbb{E}_q \|f(\mathbf{x}_s(\xi, q)) - f(\mathbf{x})\|_2 \leq \epsilon\right) - g^\top \xi\right)^2 + \lambda \|g\|_0$$

This above objective is minimized when $g = \mathbf{m}^*$, i.e., the ideal $\epsilon\mathcal{Q}$-FA mask, because this sets the first term to be zero by definition, and the second sparsity term ensures the minimality of the mask. $\square$

## B. Additional Results

**Model Verifiability.** We further test the verifiability of our model by evaluating how the model's performance changes when performing the pixel perturbation test on groundtruth attributions. This enables us to disentangle the verifiability of the model from the correctness of the attributions, as we know that our attributions are correct. We use the same groundtruth attributions as in 5.1. We report the $\ell_1$ norm between predictions made on the original samples and predictions made on the masked samples. We compare our verifiable models to the baseline models they approximate, as well as models trained with input dropout, which (Jethani et al., 2021) proposes as their verifiable model class. Training with input dropout is equivalent to training $f_v$ with random masks and cross-entropy loss rather than optimized masks and $f_b$ prediction matching. Results are shown in 3, where we see that our model performs similarly for masked and normal samples, whereas the other models do not.

|  | Hard MNIST | Chest X-ray |
|---|---|---|
| VerT Model ($f_v$) | **0.027** | **0.0009** |
| Original Model ($f_b$) | 0.107 | 0.0032 |
| $f_b$ + Input Dropout | 0.167 | 0.0536 |

Table 3: Model Verifiability

**Robustness to Explanation Attacks.** We report additional results on Chest X-ray and CelebA for the pixel perturbation tests, IOU tests, and model faithfulness tests for baseline models trained with manipulated gradients, as outlined in 5.4. We see that VerT models are still highly faithful and produce correct explanations even when derived from models adversarially trained to have manipulated explanations.

Table 4: Faithfulness of VerT Model for Manipulated Models

|  | Hard MNIST | Chest X-ray | CelebA |
|---|---|---|---|
| Accuracy on Original Data $f_v(D_d) = f_b(D_d)$ | 0.990 | 1.00 | 0.970 |
| Accuracy on Simplified Data $f_v(D_s) = f_b(D_d)$ | 0.980 | 1.00 | 0.946 |

Table 5: Intersection Over Union Results for Manipulated Models

|  |  | MNIST (manipulated) | Chest X-ray (manipulated) |
|---|---|---|---|
| Method | VerT (Ours) | **0.454** ±0.08 | **0.631**±0.12 |
|  | SmoothGrad | 0.158 ±0.07 | 0.000 ±0.00 |
|  | GradCAM | 0.040±0.06 | 0.000 ±0.00 |
|  | Input Grad | 0.002±0.01 | 0.000 ±0.00 |
|  | FullGrad | 0.333 ±0.12 | 0.004 ±0.04 |

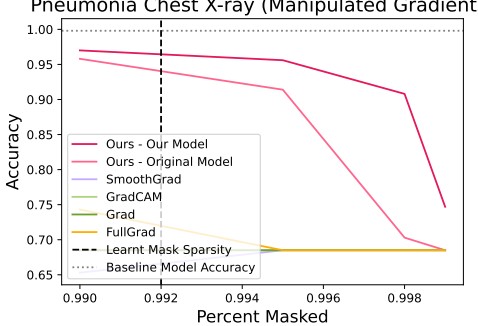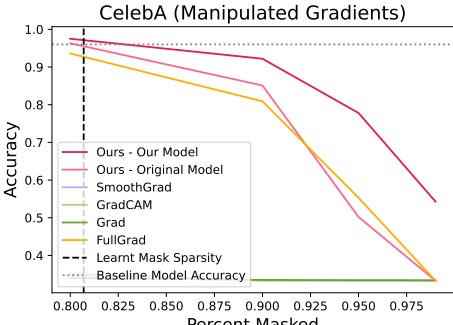

Figure 6: Pixel perturbation tests for models trained with gradient manipulation for Chest X-ray (middle) and CelebA (right).

**Attribution Sensitivity to Hyperparamaters.** We report additional results on CelebA for the ablation study on mask downscaling, as outlined in 5.5. We find minimal upscaling at higher levels of masking allows for better optimization and yields better results.

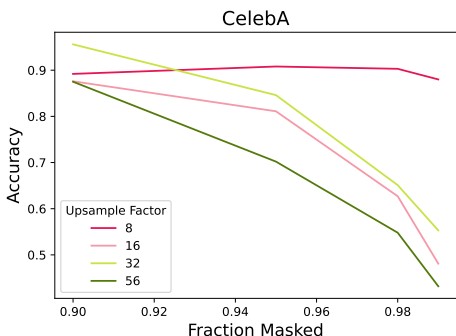

Figure 7: Ablation study of pixel perturbation test with varying levels of mask downscaling for CelebA. Example images shown in main paper.

## C. Additional Implementation and Computation Details

Models were trained on the original train/test split given by https://github.com/jayaneetha/colorized-MNIST for Hard MNIST and (Kermany et al., 2018) for the Chest X-ray dataset and with a random 80/20 split for CelebA. Baseline models were trained with Adam for 10 epochs with learning rate $1e-4$ and batch size 256. All hyperparameters are included in the code for this paper. We learn our masks with SGD (lr=300, batch size = 128 and our verifiable models with Adam (lr=$1e-4$, batch size = 128). We ran all experiments on a single A100 80 GB GPU with 32 GB memory.

## D. Broader Impact

Our method, VerT, aims to transform black-box models into verifiably interpretable models, which produce easily verifiable feature attributions. As such, if applied correctly, it can help users and stakeholders of machine learning models better understand a model's predictions and behavior by isolating the features necessary for each prediction, which can help highlight biases, overfitting, mistakes, and more. It can also help to identify spurious correlations that naturally exist in datasets and are leveraged by models through identification of the signal-distractor decomposition. However, even if VerT does not identify a spurious correlation, that does not mean that further dataset cleaning, processing, or curation is not needed, as a different model may still learn a spurious correlation that was not leveraged by the original model. Furthermore, feature attributions often do not constitute a *complete* explanation of a model. For instance, while an attribution tells us *what* was important, it does not tell us *how* it was important or how the model uses that feature. In all high stakes applications, it is still imperative that stakeholders think critically about each prediction and explanation, rather than blindly trusting either.

