# OpenReview forum: "Verifiable Feature Attributions: A Bridge between Post Hoc Explainability and Inherent Interpretability"
_ICML.cc/2023/Workshop/IMLH — IMLH 2023 Poster_

### Official Review · Reviewer_csJV · 2023-06-06
**Novel combination of post-hoc explainability with interpretable models through data adaptation, but practicality and applicability to more complex real-world scenarios are uncertain.**

**Rating:** 7
**Confidence:** 4

**Review:**

### Summary of the Paper (Long Paper):
The paper presents Verifiability Tuning (VerT) which adapts the original dataset and black-box model into an interpretable model with intrinsic verifiability due to the transformed training dataset. It motivates the presented method by introducing a formal theoretical framework for verifiability based on counterfactual reasoning and signal-distractor decomposition and empirically proves its effectiveness through various experiments on semi-synthetic and real world datasets.

### Quality:
The combination of post-hoc explainable methods and interpretable models through adapting the training data. The argumentation via signal-distractor decomposition of the training data and the usage of counterfactual reasoning to mitigate out-of-distribution observations is also conclusive. While only one evaluation method (pixel flipping) which evaluates model faithfulness (ignoring the evaluation of e.g. robustness, locality or complexity) was applied, the resulting experiments and ablation studies are interesting and convincing.

The transferability of the method to more complex problems is, however, still open to me. Especially, if the superpixel scale mask also works for more fine-grade and distributed predictive features in the image such as metastasis in a lung CT scan. Further, I wonder about the operational aspects. If I understand correctly a second model has to be trained on the new QFA loss function and a stepwise perturbed version of the original training dataset. I would assume that this can take up a lot of time and resources for models which are originally trained on large scales of training instances over several days, compared to almost instantly applied post-hoc XAI methods such as the ones compared to in this paper.

### Clarity:
The paper is very well written and the plots accompanying the experiments illustrate the results very well. For sections 5.4 and 5.5 however, only the results for HardMNIST are shown (and one CelebA image in Figure 5). I think it would have been beneficial for the paper to also add the plots for the other two datasets to the Appendix, as showing the results only for the synthetic dataset leaves the reader in uncertainty about the results on more real world datasets.

### Originality:
The paper acknowledges the similarity of their approach to JAM and shows the distinctive difference of their approach. The first point the paper makes is that it “trains models from scratch, whereas VerT can interpret black-box models” I find only partly convincing from an operational perspective because also in VerT a surrogate model has to be trained. In this sense a comparison to JAM would have been interesting also with regard to the ablation studies. Otherwise the paper is convincing with its motivation and interesting experiments.

### Significance:
The paper tackles an unsolved problem of verifiability in the XAI domain. While the results seem convincing, the real-world applicability of the method regarding operational factors and more complex features in the data is uncertain.

Minor remarks:
- The reference to equation 2 is not working in line 218 (second column).

---

### Official Review · Reviewer_64wR · 2023-06-07
**Tuning Nerual Networks to Better Reflect Feature Attributions**

**Rating:** 7
**Confidence:** 3

**Review:**

### Summary

This is a long paper submission (8 pages). This paper studies feature attribution in neural network models; in particular, (i) it tries to formulate what it means for a feature attribution to be "verifiable" and (ii) introduces a tuning approach to align models with their attributions better.

### Strengths

* To my knowledge, the ideas presented in this paper are novel and will be, in my opinion, interesting to the community: the paper proposes a framework to characterise models and feature attributions and, in addition, describes a practical approach for tuning neural networks to align better with their attributions.

* The proposed method and relevant baselines are tested on various image datasets, including benchmarks where the ground-truth attributions are known.

* Overall, the ideas are presented and explained clearly, making the paper more accessible to the community.

### Weaknesses

* As pointed out by the authors, the whole framework and the tuning technique assume that the image is somehow decomposable into spatially separated regions containing "signal" and "distractor". Hence, like many other attribution methods, VerT would not produce very meaningful explanations for classifiers relying on hue or texture, for example.

* A few assumptions and design choices remain unexplored. For instance, in the experiments, $\mathcal{Q}$ is fixed to a normal distribution centred around the mean pixel value. It would be interesting to see some experiments on how VerT behaves under different choices of $\mathcal{Q}$. Moreover, it would be great to provide some discussion/guidance on choosing $\mathcal{Q}$ in practice. Similarly, some choices in the definition of the QFA have not been discussed too much. For example, why must the attribution be sparse (minimising the $\ell_0$-norm, lines 160-164)? Isn't it a relatively strong assumption on the character of the desired attribution? Isn't it also somewhat restrictive that the attribution must be a binary mask, i.e. a feature is strictly important or not?

* I wonder if any related works on interpretable models propose similar training procedures but do not necessarily motivate the method from the attribution perspective. For instance, it would be interesting to discuss the relationship between the proposed framework and (deep) attribution priors [1,2], which similarly try incorporating attribution into the training procedure. Another attribution technique that seems to build on feature masking and raises similar ideas is CXPlain [3].

### Minor Points/Further Remarks/Questions

* In Equations 1 and 2, shouldn't the expectation also be w.r.t. $q\sim\mathcal{Q}$?

* Section 2 contains many unexplained abbreviations (GLMs, GAMs, JAMs).

* Section 2 claims that "the performance of interpretable models often suffers when compared to unconstrained black-box architectures." It would be good to include some references here to support the presence of the tradeoff. I think, in many datasets, it is very debatable whether such a tradeoff even exists.

* Line 59, p. 2, second column: "While (Chen et al., 2019; Koh et al., 2020) leverage the expressivity of deep networks..."---this is an incorrect use of the APA citation style. In this case, the citation should be replaced by "Chen et al. (2019) and Koh et al. (2020)" (e.g. see https://apastyle.apa.org/style-grammar-guidelines/citations/basic-principles/parenthetical-versus-narrative).

### References
[1] Erion, G., Janizek, J. D., Sturmfels, P., Lundberg, S. M., & Lee, S. I. (2021). Improving performance of deep learning models with axiomatic attribution priors and expected gradients. *Nature machine intelligence, 3*(7), 620-631.

[2] Weinberger, E., Janizek, J., & Lee, S. I. (2020). Learning deep attribution priors based on prior knowledge. *Advances in Neural Information Processing Systems, 33*, 14034-14045.

[3] Schwab, P., & Karlen, W. (2019). CXPlain: Causal explanations for model interpretation under uncertainty. *Advances in Neural Information Processing Systems, 32*.

---

### Official Review · Reviewer_iev4 · 2023-06-17
**Good paper, should be accepted**

**Rating:** 7
**Confidence:** 3

**Review:**

The authors present a new strategy for training an image classifier that yields interpretable masks.

Their approach is interesting, and their evaluation is reasonable, so it should be accepted.

However, there's some weaknesses:
- Table 1 is key, but I don't think the results there are too convincing - in fact they are perhaps a bit misleading:
-- For HardMNIST, their method OUGHT to win, since it has a super-pixel bias and (to me) it looks like the "ground truth" attributions are very well covered by a superpixel.
-- For Chest X-ray (really just a simple dataset with injected shortcut signal), I suspect GradCAM performs poorly because the artefact in question is at the very top of the image.This is the case for all 3 examples shown and thus perhaps for all examples? (The description of the dataset doesn't specify).
- There is missing discussion on the performance details of their method. The model for (ours) is not the same as that for the other methods, and requiring changes to the model is both a strength and a potential weakness for this method. In the future, the authors should show more detailed experiments of model performance/training cost/etc. for their model vs standard resnets.

---

### Meta-Review · Area_Chair_8oMT · 2023-06-19

**Recommendation:** Accept (Poster)
**Confidence:** 4

**Metareview:**

This work proposes the Verifiability Tuning method to fine-tune a black-box model on verifiable feature attributes. The method and experiment is solid. The authors should address the reviewers comments accordingly to make the manuscript more solid.

---

### Decision · Program_Chairs · 2023-06-20

Accept (Poster)